



# High-resolution multi-scaling of outdoor human thermal comfort and its intra-urban variability based on machine learning

Ferdinand Briegel[1], Jonas Wehrle[1], Dirk Schindler[1], Andreas Christen[1]

[1]Chair of Environmental Meteorology, Faculty of Environment and Natural Resources, University of Freiburg, Freiburg im Breisgau, Germany

*Corresponding author*: Ferdinand Briegel (ferdinand.briegel@meteo.uni-freiburg.de)

**Abstract.** As the frequency and intensity of heat waves will continue to increase in the future, accurate and high-resolution mapping and forecasting of human outdoor thermal comfort in urban environments is of great importance. This study presents a machine learning based outdoor thermal comfort model with a good trade-off between computational cost, complexity, and accuracy compared to common numerical urban climate models. The machine learning approach is basically an emulation of different numerical urban climate models. The final model consists of four sub-models that predict air temperature, relative humidity, wind speed, and mean radiant temperature based on meteorological forcing and geospatial data on building form, land cover, and vegetation. These variables are then combined into a thermal index (Universal Thermal Climate Index - UTCI). All four sub-model predictions and the final model output are evaluated using street-level measurements from a dense urban sensor network in Freiburg, Germany. The final model has a mean absolute error of 2.3 K. Based on a city-wide simulation for the city of Freiburg we demonstrate that the model is fast and versatile enough to simulate multiple years at hourly timesteps to predict street-level UTCI at 1 m spatial resolution for an entire city. Simulations indicate that neighborhood-averaged thermal comfort conditions vary widely between neighborhoods, even if they are attributed to the same local climate zones, e.g. due to differences in age and degree of urban vegetation. Simulations also show contrasting differences in the location of hot spots during the day and at night.

## 1    Introduction

The frequency and severity of heatwaves have increased and are expected to increase even further due to human-caused climate change (IPCC, 2021). In addition, heatwaves are occurring earlier in summer, resulting in a longer period of potential heat stress (IPCC, 2021). Rousi et al. (2022) found that the frequency and intensity of heatwaves increased 3–4 times faster in Europe than in the rest of the mid-latitudes over the past decades. In 2020, heatwaves in Western Europe accounted for 42 % of all reported global deaths from extreme weather events, with a total of 6340 deaths (CRED, 2021). As the severity of heatwaves also depends on land cover and land use, urban areas are even more exposed to extreme heatwave events than rural areas due to their physical characteristics (Unger et al., 2020; Masson et al., 2020), including reduced nocturnal cooling and limited access to cool microenvironments for urban populations.





Human thermal comfort is influenced not only by air temperature ($T_a$) but also by wind speed ($U$), radiation, and humidity. The variables expressing the effect of radiation and humidity are the mean radiant temperature ($T_{mrt}$) and relative humidity (RH), respectively. Thermal indices combine these four environmental variables to describe thermal comfort and overall thermal stress of an individual (Epstein and Moran, 2006). Multiple thermal indices have been developed, such as the Physiological Equivalent Temperature (PET) or Universal Thermal Climate Index (UTCI) (Coccolo et al., 2016; Potchter et

al., 2018). Several studies have concluded that $T_{mrt}$ is the driver of outdoor human thermal comfort during daytime (Cohen et al., 2012; Holst and Mayer, 2011; Kántor and Unger, 2011). In addition to terrain, the complex and heterogeneous three-dimensional structure of cities causes high spatial and temporal variability of these environmental variables. $T_{mrt}$ and $U$ have the highest variability at the micro-scale (Matzarakis et al., 2016). However, $T_a$ also varies locally, although less strongly (Fenner et al., 2017; Quanz et al., 2018; Shreevastava et al., 2021). In particular, $T_a$ is more relevant to human thermal comfort

during nighttime than $T_{mrt}$ (Lee et al., 2013). As this study is concerned with a city-wide multi-scale approach to modeling outdoor human thermal comfort, the focus is on modeling $T_a$, RH, $U$, and $T_{mrt}$ within the urban canopy layer (at about 1.1– 2.0 m a.g.l.) at the neighborhood scale ($T_a$ and RH: 500 x 500 m) and building-resolved scale ($T_{mrt}$ and $U$: 1 x 1 m).

In recent years, several deterministic and stochastic modeling approaches have been developed to map urban climate, outdoor thermal comfort, and canopy urban heat island (UHI) at different scales and layers and with varying complexity (Mirzaei,

2015). Mesoscale models have been parametrized for urban surfaces, the so-called slab or bulk models (Dupont et al., 2006), and coupled with urban canopy models (Chen et al.,2011; Hamdi et al. 2012; Martilli et al., 2002; Masson, 2000; Rafael et al., 2020). Urban canopy models, on the other hand, have also been used as stand-alone (offline) models to investigate the urban surface energy and water balance at the local scale (Best and Grimmond, 2015, and Grimmond et al., 2011). In addition to numerical urban climate models, statistical models (Chen et al., 2022; Ho et al., 2014; Straub et al., 2019) and dense

measurement networks (Gubler et al., 2021) have been used to map urban climate. It can be concluded that there are several ways to model urban climate on various scales. However, as pointed out by Hamdi et al. (2020) and Masson et al. (2020), the complexity of the model should depend on the application and only be as complex as necessary.

Recently, machine and deep learning models (ML) have gained increasing attention in urban meteorology and have been used to emulate numerical urban climate models (Meyer et al., 2022; Briegel et al., 2023). The micro-scale $T_{mrt}$ model

SOLWEIG (SOlar and LongWave Environmental Irradiance Geometry – Lindberg and Grimmond, (2019)) was emulated by a deep convolutional encoder-decoder approach, called U-Net, which showed a promising trade-off between accuracy and computational cost (Briegel et al., 2023). Meyer et al. (2022) emulated an ensemble of Urban Land Surface Models (ULSM) using a simple Multi-Layer-Perceptron (MLP). The MLP was used to model energy and radiation fluxes and was compared to a reference ULSM (Town Energy Balance model). The MLP was found to be more accurate and more stable, especially in

online mode. It can be concluded from these studies that the advantage of ML models lies in their lower computational cost once trained. Nevertheless, an emulated ML model can never exceed the accuracy of a numerical model, as it is trained on its model results.





This study proposes a novel and fast computational ML approach to model outdoor human thermal comfort at 1 x 1 m resolution in complex urban areas, hereafter called the Human Thermal Comfort Neural Network (HTC-NN). The HTC-NN
can be used to downscale numerical weather prediction models or reanalysis data, considering urban geometry and function, and predict outdoor human thermal comfort at high resolution at a limited computational cost. The HTC-NN consists of four submodels: two neighborhood scale MLP models for modeling $T_a$ and RH; a building-resolved U-Net model for modeling $T_{mrt}$; and a building-resolved statistical wind field. We use the MLPs to emulate a surface energy balance model to model $T_a$ and RH at the neighborhood scale (500 x 500 m) at 2.0 m a.g.l. and a U-Net to emulate a $T_{mrt}$ model at building-resolved scale
(1 x 1 m) at 1.1 m a.g.l. (Briegel et al., 2023). In addition, Large-Eddy Simulations (LES) are emulated by a random forest (RF) model to compute statistical wind fields at 1.0 m a.g.l. in the urban canopy layer with a 1 x 1 m resolution for the four cardinal wind directions in relation to the forcing data. The single-layer model Surface Urban Energy and Water balance Scheme (SUEWS) is used as surface energy model for the neighborhood scale modeling of $T_a$ and RH (Järvi et al., 2011; Ward et al., 2016; Sun and Grimmond, 2019). SOLWEIG is the building-resolved $T_{mrt}$ model (Lindberg and Grimmond, 2019). The
U-Net that emulates SOLWEIG has already been developed and validated (Briegel et al., 2023). LES are used to obtain the scaling matrices and maps of mean wind speed in relation to forcing wind speed based on the LES of Albertson and Parlange (1999a, 1999b).

The objectives of this study are (i) to develop and validate ML models emulating numerical urban climate models to predict $T_a$, RH, and $U$ at different scales, (ii) to link these models to predict and validate spatially distributed UTCI at 1 x 1 m resolution,
and (iii) to map UTCI at high resolution in a case study of an entire urbanized area (Freiburg, Germany) over many years to derive a climatology of intra-urban variability of outdoor thermal climate.

## 2 Methods

### 2.1 Study area

The study area covers the urbanized area of Freiburg im Breisgau, Germany, and is partially identical to the study area reported
in Briegel et al. (2023). According to the Local Climate Zone (LCZ) classification, the urbanized area of Freiburg can be mainly classified into LCZs 5 (open midrise), 6 (open lowrise), and 8 (large lowrise), while some parts of the city center are LCZs 2 (compact midrise) and 3 (compact lowrise) (Stewart and Oke, 2012; Demuzere et al., 2022). Due to different training requirements, the training and testing model domains for the individual submodels differ, as well as the HTC-NN prediction domain. The MLP submodel training domains have an extent of 15 x 15 km and a grid size of 500 m, resulting in 436 grid
cells. The training domain of the MLPs is larger than the actual HTC-NN prediction domain. The RF $U$ submodel training domain covers 15 areas of varying grid size ranging from 122 x 122 m to 500 x 500 m. The final HTC-NN model prediction domain has an extent of 10 x 7 km and a grid size of 1 x 1 m. An overview of the training and prediction domains, individual training and test areas, and the urban sensor network is given in Fig. 1.





## 2.2 Spatial and forcing data

Spatial and forcing data are largely similar for the numerical and their corresponding ML submodels, however, the MLPs use only a subset of all SUEWS input data (see Sect. 2.4), and the RF uses additional spatial predictors. The derivation of all spatial data other than anthropogenic dynamics is described in Briegel et al. (2023). A detailed overview of the spatial input data required for each submodel and their derivation is given in Table 2.

    SUEWS does not require explicit building-resolving data but local-scale averaged spatial data. Information on urban

morphology, land cover classes, and anthropogenic dynamics are required for SUEWS. Population density for each grid cell is derived from demographic data by city district (City of Freiburg im Breisgau - Bevölkerung, 2022). The remaining spatial input data, such as emissivity and albedo, are left at default (Sun et al., 2021; Sun and Grimmond, 2019).

    LES and SOLWEIG require three-dimensional building geometry and tree characteristics derived from digital surface models (DSMs), digital elevation models (DEMs), and building outline data (Briegel et al., 2023), hereafter referred to as DSMb and

DSMv.

    Measurements from the urban weather station of the University of Freiburg are used as meteorological forcing data. The weather station is located on a rooftop (~ 55 m a.g.l.; 48.0011, 7.8486) close to the city center (Fig. 1). A detailed description of the urban weather station can be found in Briegel et al. (2023). The following variables are used as forcing data for SUEWS, the corresponding MLPs, and the U-Net: $T_a$, RH, atmospheric pressure, downwelling shortwave radiation, downwelling

longwave radiation, precipitation, $U$ and wind direction. LES requires only standard forcing related to an initial shear velocity of 1 m s$^{-1}$.

## 2.3 Validation Data

Validation data for the MLPs and HTC-NN are derived from an urban weather sensor network in Freiburg (Fig. 1), which was installed in the summer of 2022. The sensor network covers the entire urbanized area as well as some rural areas, adjacent

valleys, and hills to account for local weather phenomena such as mountain-valley wind systems or altitude effects which are not resolved by the ML model.

    The stations of the sensor network can be divided into Tier I stations, or biometeorological stations (7 stations), and Tier II stations (30 stations). Tier I stations measure $T_a$, precipitation, RH, wind speed and direction, global radiation, and black globe temperature (Feigel et al., 2023), while Tier II stations measure only $T_a$, precipitation, and RH (Plein et al., 2023). Tier I stations

are equipped with full weather sensors (ClimaVUE50, Campbell Scientific Inc., Logan, UT, USA) and black globe temperature sensors (model BLACKGLOBE-L, Campbell Scientific Inc., Logan, UT, USA). All sensors are mounted at 3.5 m a.g.l. on streetlights or custom poles, and the measuring interval is one minute. Sensor network data in this study is aggregated to hourly values.





**Table 1: Overview of the abstract input data, the spatial data and the method used to derive the abstract input data. Most calculations are performed by the Urban Multi-scale Environmental Predictor (UMEP - Lindberg et al. (2018)). DEM; DSM; b: buildings; v: vegetation.**

| Abstract spatial input | Spatial data | Method |
|---|---|---|
| Surface cover fractions | LCC map | UMEP Land Cover Fraction |
| Zero displacement height (b / v) | DEM, DSMb / DSMv | UMEP Morphometric Calculator: (Kanda et al., 2007) |
| Roughness length (b / v) | DEM, DSMb / DSMv | UMEP Morphometric Calculator: (Kanda et al., 2007) |
| Mean height (b / v) | DEM, DSMb / DSMv | UMEP Morphometric Calculator: (Kanda et al., 2007) |
| Frontal area index (b / v) | DEM, DSMb / DSMv | UMEP Morphometric Calculator: (Kanda et al., 2007) |
| Altitude | DEM | Mean height of grid cell |
| Population density | Population data by district | Weighted mean of different districts in specific grid cell |

Besides the sensor network, SOLWEIG is run for small subsets (50 x 50 m) around the Tier I stations to derive UTCI and $T_{mrt}$ data for the specific locations of the Tier I stations (POI function - Lindberg and Grimmond (2019)). This allows a more detailed evaluation and a better attribution of the HTC-NN results.

## 2.4 Study period

The study period is aligned with the sensor network data which are collected from June 2022 onwards. The entire model period is from 2018 to 2022, with 2018 used as a spin-up year for SUEWS. The MLPs are trained with data from 2019 to 2021 and tested for 2022 (January–December). Model validation of SUEWS and the MLPs are performed with measurement data from June to December 2022, while the HTC-NN is validated with data from August to December 2022.

## 2.5 Modeling approach

The development of the HTC-NN requires four steps (Fig. 2). The first step is to generate initial spatial and meteorological data from various sources. In the second step, the so-called 'ground truth' data ($T_a$, RH, $T_{mrt}$, and $U$) for the four HTC-NN submodels (two MLPs, U-Net, and RF) are calculated using numerical models. Training and evaluation of the submodels are done in the third step, while the fourth step is to link these submodels by calculating UTCI. As the U-Net has already been trained and validated, only the development and the requirements of the MLPs and the RF (spatial and temporal data, SUEWS, and LES) are explained.



**Figure 1: Model domain of the City of Freiburg, Germany. The red star shows the location of the weather station used for forcing data on a rooftop. Orange and turquoise points show the locations of the urban sensor network used for model evaluation. Gray grid cells show the training areas of the $T_a$ and RH submodels, while yellow grids show the test areas. Red and blue squares show the training and test areas of the $U$ sub model, respectively. The pink border shows the prediction area of UTCI. Orthophoto in the background based on data from the City of Freiburg, www.freiburg.de.**

### 2.5.1 Local scale $T_a$ and RH modeling

SUEWS (version 2020a) is used to model $T_a$ and RH at 2.0 m a.g.l. for 436 grid cells with a resolution of 500 x 500 m (Järvi et al., 2011; Ward et al., 2016). SUEWS has been validated in different cities under different climatic conditions (Ward et al., 2016; Järvi et al., 2011; Ao et al., 2018). Besides the following parameters, SUEWS is run in default mode: net radiation method, maximum and minimum porosity, roughness length of momentum method. Netradiation method is set to 1, as downwelling longwave radiation data are available. Maximum and minimum porosity of deciduous trees is set to 0.6 and 0.2, respectively (Ward et al., 2013). Roughness length and zero displacement height are calculated according to Kanda et al.





(2007) and are provided to SUEWS, so the roughness length method is set to 1. As mentioned, SUEWS is run for 2018–2022, while 2018 is excluded from subsequent modeling as it serves as a spin-up year.

**Table 2: Overview of required spatial data for the numerical and machine learning models. Note: SUEWS and MLP use abstract spatial data (see Table 1) and the RF model uses additional spatial predictors derived from DEM, DSMb, and DSMv which are not listed here.**

| Data | SUEWS / MLP (500 x 500 m) | SOLWEIG / U-Net (1 x 1 m) | LES / RF (1 x 1 m) |
|---|---|---|---|
| LCC map | x | x | - |
| DEM | x | x | x |
| DSMb | x | x | x |
| DSMv | x | x | x |
| Sky view factor | - | x | - |
| Wall height and aspect | - | x | - |
| Soil characteristics | x | - | - |

### 2.5.2 Micro-scale $U$ modeling

LES are used to obtain micro-scale (1 x 1 m) wind fields for 15 areas ranging in size from 122 x 122 m to 500 x 500 m. The LES model code and set-up are described in Giometto et al. (2016) and (2017). LES needs DSMb and DSMv as spatial input data. Training and test areas are carefully selected to ensure that the variability of the urban environment is adequately represented. For each area, four LES are computed for pressure gradients that cause north, east, south, and west inflow directions. The simulations are set up with an identical standard forcing related to an initial shear velocity of 1 m s$^{-1}$. Each simulation contains 30 minutes (steady-state) wind field data in two-second steps, which are then time-averaged.

### 2.5.3 Multi-layer perceptron model development

The MLPs are designed using Python and the PyTorch library. To determine the best model architecture, a model-based Bayesian hyperparameter optimization is performed. For this purpose, the Optuna software framework (Akiba et al., 2019) is used in combination with PyTorch. Training and prediction are conducted on an NVIDIA GeForce RTX 3080.

$T_a$ and RH are each modeled with their own deep learning model. Different to the $T_{mrt}$ deep learning model, which is based on a convolutional neural network, $T_a$ and RH models are built from fully-connected feedforward artificial neural networks, known as MLP. This is because the different SUEWS grid cells are not connected and have no spatial relationship. MLPs consist of three different types of layers: input layer, hidden layers, and output layer. Bayesian optimization determines the





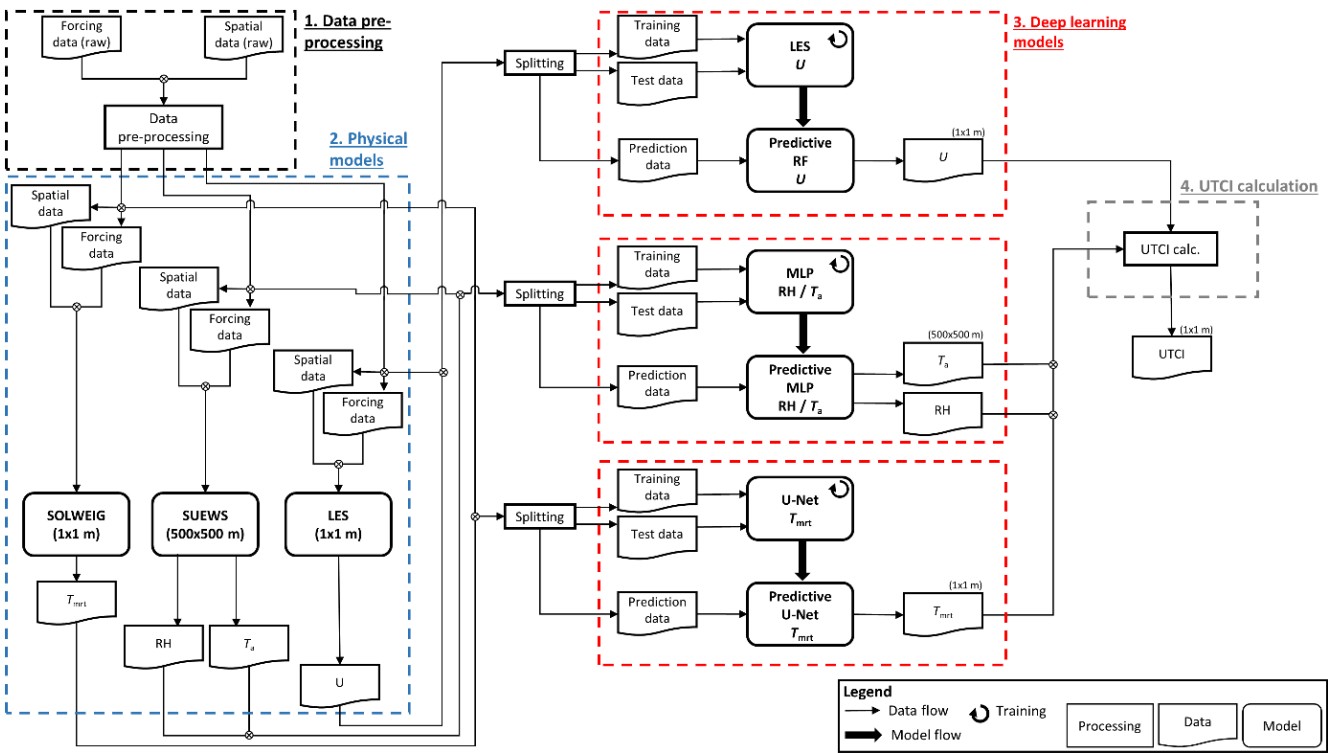

**Figure 2: Workflow of the HTC-NN model development: 1. Data pre-processing; 2. Numerical modeling of 'ground truth'; 3. Training and evaluation of machine learning models; and 4. Calculation of UTCI.**

number of hidden layers and neurons, the presence/absence of a dropout layer, and the learning rate that leads to the highest

model accuracy. In total, 30 hyperparameter combinations (trials) are tested for each model. For this purpose, the training dataset is further divided into training and evaluation datasets, and a 3-fold cross-validation is used to validate each hyperparameter trial. The hyperparameter combination of each trial is defined by the Tree-structured Parzen Estimator (TPE) (Bergstra et al., 2013, 2011), which is a Bayesian hyperparameter optimization algorithm. The best architectures of both MLPs have three hidden layers with 60, 49, and 42 neurons for the $T_a$ MLP and 34, 53, and 54 neurons for the RH MLP. Learning

rates are best at 0.0014 and 0.0011 for the $T_a$ MLP and RH MLP, respectively. Drop-out layers do not improve the model accuracy and are not added to the final models. The remaining hyperparameters, such as activation function, optimizer, and initial weight distribution, are taken from the literature (Table 3 and Briegel et al. (2020)). As an evaluation metric (loss function), mean squared error (MSE) is used. For further comparisons, root mean square error (RMSE) as well as mean bias error (MBE) are used.

**2.5.4    Random Forest model development ($U$)**

The RF modeling is conducted in MATLAB (The Mathworks, Natick, MA, USA, version 2021b) using a regression-bagging approach with 50 learning cycles. In addition to spatial building and tree data, spatial features (derived from DSMb) are





predictors. These predictors are indices pertaining to the street length and width, frontal area index relative to the flow direction, horizontal Euclidean distance, and up- and downwind distances to buildings. For each of the 12 training areas and the three

components of the wind field, individual models are trained (*u*: streamwise, *v*: normal-to-streamwise horizontal, *w*: vertical), resulting in 36 models. This model ensemble makes predictions for each component, which are then assembled as the Euclidean norm of *u*, *v*, *w* to obtain the final wind field. The RF models can theoretically compute wind fields for any wind direction over the city, requiring only an adjustment of the predictors. However, for the purpose of calculating outdoor thermal comfort, citywide wind fields are computed only for the north (315–45 °), east (45–135 °), south (135–225 °), and west (225–315 °)

inflow directions, each covering a 90° angle. The LES wind fields are scale-independent, allowing a linear rescaling of the RF model output. Thus, a time series of city-wide wind fields can be computed by hourly wind speed data from the urban weather station.

## 2.6 Thermal Indices

In the final model step, the results of the above-mentioned submodels are combined into a thermal comfort index with a spatial

resolution of 1 x 1 m. However, not all indices are appropriate for human thermal comfort (Staiger et al., 2019). Therefore, only UTCI (Błażejczyk et al., 2013), which is widely used in urban climate science and planning, is considered in this study. The reference conditions of UTCI correspond to an individual walking outdoors with $T_{mrt}$ equal to $T_a$, no wind and RH at 50 %.

**Table 3: Hyper-parameters of both MLP models.**

| Hyper-parameter | Value |
|---|---|
| Activation function | ReLU |
| Optimizer | Adam |
| Initial weight distributions | He uniform |
| Loss function | MSE |
| Epochs | 20 |
| Batch size | 62 |
| Hidden layers | $T_a$ MLP: 3 |
| | RH MLP: 3 |
| Neurons | $T_a$ MLP: 60, 49, 42 |
| | RH MLP: 34, 53, 54 |
| Learning rate | $T_a$ MLP: 0.0014 |
| | RH MLP: 0.0011 |





## 3 Results

This section presents the evaluation of the three submodels of the HTC-NN (Sect. 3.1), of the HTC-NN itself (Sect. 3.2) and the high-resolution UCTI mapping of the HTC-NN (Sect. 3.3).

### 3.1 Evaluation of $T_a$, RH, and $U$ submodels

The accuracies of SUEWS, SOLWEIG, the MLPs, U-Net, and RF relative to the sensor network are shown in Fig. 3 and Table 4. A spatial evaluation of machine learning models to numerical models on their specific test areas is given in Table A1.

With an RMSE of 1.50 K and 1.51 K respectively, the SUEWS and $T_a$ MLP models have similar performance compared to the sensor network data. Both models have a lower RMSE of about 0.3 K compared to the forcing data. The error distributions of SUEWS and the MLP across the different stations are similar (Fig. 3a), with a higher variability during the night than the

during day. Figure 4b shows a moving average of RMSE over time of the final HTC-NN model and of the $T_a$ and RH MLPs (Fig. 4c and d). It can be seen that the RMSE of the $T_a$ MLP shows two peaks in late October and mid-December, while the RMSE fluctuates around 1 K during the remaining time. Overall, the $T_a$ MLP shows good accuracy to the SUEWS model ($R^2$ of 0.997 and RMSE of 0.5 K). Similar observations can be made for the RH MLP. RMSE for SUEWS and the MLP model are 7.79 % and 8.14 %, respectively, while the RMSE of the forcing data is 9.08 %. Similar to the $T_a$ models, the RMSE of the

RH MLP has two strong peaks in late October and mid-December with RMSE values up to 15 %, while RMSE fluctuates

**Table 4: MAE, MBE, RMSE, and coefficient of determination ($R^2$) of numerical / machine learning models and sensor network data for $T_a$, RH, $U$, $T_{mrt}$, and UTCI. Additionally, errors between forcing data and sensor network data are added for $T_a$, RH, and $U$ (as baseline). Note, $T_a$ and RH are validated on Tier I and Tier II stations, while $U$, $T_{mrt}$, and UTCI are only validated on Tier I stations.**

| Variable | Model / Data | MAE | MBE | RMSE | $R^2$ |
|---|---|---|---|---|---|
| $T_a$ | SUEWS | 1.07 K | -0.15 K | 1.50 K | 0.97 |
| | MLP | 1.08 K | -0.17 K | 1.51 K | 0.97 |
| | Forcing | 1.32 K | -0.12 K | 1.83 K | 0.95 |
| RH | SUEWS | 5.80 % | -0.44 % | 7.97 % | 0.88 |
| | MLP | 5.99 % | -0.33 % | 8.14 % | 0.87 |
| | Forcing | 6.54 % | -1.52 % | 9.08 % | 0.84 |
| $U$ | RF | 0.52 m s$^{-1}$ | 0.24 m s$^{-1}$ | 0.73 m s$^{-1}$ | 0.28 |
| | Forcing | 2.27 m s$^{-1}$ | -2.24 m s$^{-1}$ | 2.85 m s$^{-1}$ | 0.35 |
| $T_{mrt}$ | U-Net | 4.25 K | -1.53 K | 6.18 K | 0.84 |
| | SOLWEIG | 3.83 K | -0.97 K | 5.86 K | 0.86 |
| UTCI | HTC-NN | 2.27 K | 1.34 K | 3.00 K | 0.92 |
| | SOLWEIG | 2.48 K | -0.66 K | 3.29 K | 0.90 |





**Figure 3: Boxplots of RMSE of SUEWS and MLPs compared to sensor network data and between SUEWS and the MLPs for $T_a$ (a) and RH (b) respectively and their differences between day and night. In (c) RMSE of $U$ between sensor network data and RF is shown. Boxplots show dispersion of different sensor network stations.**

around 5 to 6 % the remaining time. $R^2$ of SUEWS and the RH MLP are lower at 0.88 and 0.87 compared to 0.97 of the $T_a$ models. Nevertheless, the overall $R^2$ between SUEWS and the RH MLP is high at 0.98, and with an overall RMSE of 3.28 %, the accuracy of the RH MLP is considered satisfactory. The $T_{mrt}$ U-Net has a slightly lower accuracy than SOLWEIG (RMSE of 6.18 K to 5.86 K; $R^2$ of 0.84 to 0.86). A detailed evaluation can be found in Briegel et al. (2023). The RF has an $R^2$ of 0.28 and an RMSE of 0.74 m s$^{-1}$ in relation to sensor network data and shows a large improvement to forcing data (RMSE of 2.85 m s$^{-1}$). However, $R^2$ of RF $U$ is lower than $R^2$ of forcing data. The RF model has an overall accuracy of 0.76 m s$^{-1}$ (RMSE) compared to the LES model (Table A1).





**Figure 4: Moving average of UTCI (a) and RMSE of UTCI (b), $T_a$ (c), RH (d), $T_{mrt}$ (e), and $U$ (f) from August to December 2022. The window size of the moving average is seven days. Time series starts with the installation of the first Tier I stations in August 2022. In (c) and (d) RMSE of SUEWS predictions are added as comparison (orange lines). In (b) and (e) RMSE of SOLWEIG predictions are added (violet lines). For $U$ no numerical model results exists. Shaded areas represent 95 % confidence interval.**

## 3.2   HTC-NN evaluation

The HTC-NN has an RMSE of about 3.00 K and an $R^2$ of 0.92 (in relation to sensor network), while SOLWEIG has an RMSE of 3.29 K and $R^2$ of 0.90 (Table 4). Moving averages of UTCI and RMSE (window size of 7 days) over time (August–December 2022) and diurnal error patterns are shown in Fig. 4 and 5. The annual RMSE of UTCI ranges between 2 and 4 K, with the highest RMSE values in mid-September, late October, and mid-December. $T_a$ and RH show a similar pattern in October and December, while $T_{mrt}$ has its largest errors in August and September, ranging from 6 to 8 K. The RMSE of $U$ varies between

0.5 and 1 m s$^{-1}$ with the highest values in August, late October, and late December. The elevated RMSE values of UTCI in mid-September coincide with the RMSE peak of $T_{mrt}$ in mid-September, while the RMSE peaks of UTCI in late

**Figure 5: Normalized diurnal RMSE of UTCI (a), $T_a$ (b), RH (c), $T_{mrt}$ (d) and $U$ (e). In (b) and (c) RMSE of SUEWS predictions are added as comparison (orange lines). In (a) and (d) RMSE of SOLWEIG predictions are added (violet lines). For $U$ no numerical model results exists. Shaded areas represent 95 % confidence interval.**

October and December match the high RMSE values for $T_a$ and RH in those periods. In Fig. 4b and e, results from SOLWEIG are added (violet lines). SOLWEIG is more accurate than the HTC-NN in modeling UTCI / $T_{mrt}$ in summer, while the HTC-
NN is more accurate in autumn and winter. Diurnal patterns of UTCI accuracy are shown in Fig. 5a. Model errors are lower during the night than during the day, with the highest errors during the morning. The U-Net $T_{mrt}$ error is also lowest during the night but has its highest errors in early evening (see also Briegel et al. (2023)). The $T_a$ and RH MLPs have similar error patterns and are lowest in the late afternoon. The diurnal pattern of the $U$ RF model accuracy shows higher accuracy during the day than at night.

Figure 6 shows sensor network and modeled UTCI values (HTC-NN and SOLWEIG) at five stations during an exemplary heat wave event in August 2022. Daytime UTCI is overestimated by the HTC-NN to varying degrees (between 1 and 5 °C), while nighttime predictions are in line with measurements. In Fig. 6c, the HTC-NN underestimated UTCI in the afternoon,



**Figure 6: Modeled and measured UTCI and $T_{mrt}$ at five Tier I stations (a)–(e) during a heatwave event in August 2022.**

which is in line with underestimated $T_{mrt}$. SOLWEIG, on contrary, underestimates UTCI in the morning and overestimates it

in the afternoon and evening. Figure 6e shows significantly lower UTCI and $T_{mrt}$ values of SOLWEIG during the afternoon, while the HTC-NN does not show this pattern.

In Fig. 7, distributions of both sensors and models and their differences are shown. The HTC-NN has a higher share of values between 15 and 25 °C, whereas shares of UTCI greater 25 °C are equal. SOLWEIG, on the contrary, has lower shares for almost all the bins below 2.5 °C and higher shares between 5 and 12.5 °C.


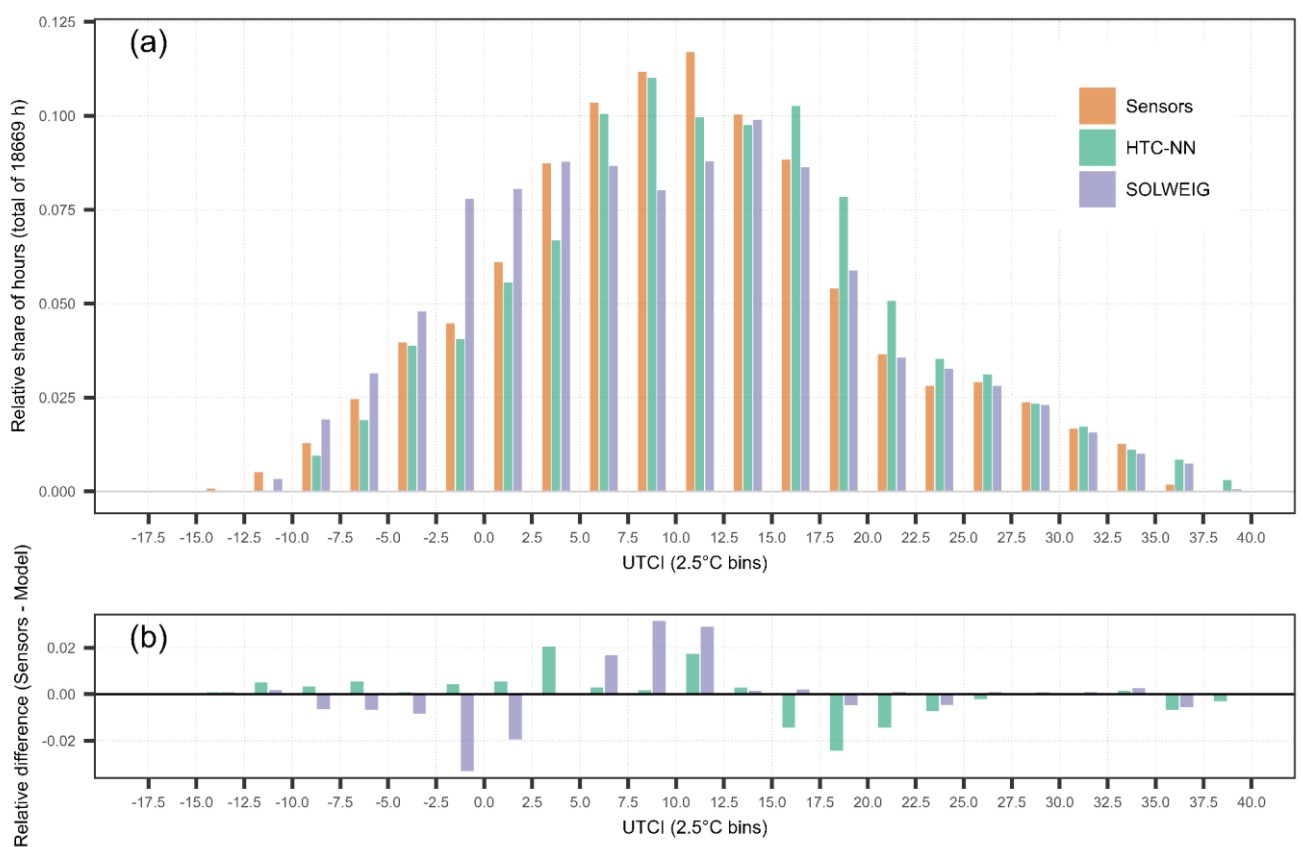

**Figure 7: UTCI distributions from sensors, HTC-NN, and SOLWEIG (a) and the differences between models and sensor network (b). Distributions are shown for bins with a size of 2.5 °C and their relative share on total hours (n = 18669 h).**

### 3.3 High-resolution UTCI mapping

This section presents an application of the developed HTC-NN for mapping and downscaling simulated UTCI at 1 m resolution for Freiburg. Calculating hourly UTCI over four years (35064 timesteps) and 42.5 million pixels using an Intel Core i9 processor and NVIDIA GeForce RTX 3080 took only about 8 days. No individual runs are saved as this would exceed storage capacities, and predictions are stored as day- and nighttime sums of hours for each month within 1 °C UTCI bins.

Figure 8 shows the spatial distribution of cumulative daytime hours with strong, very strong, or extreme heat stress. This

figure shows heat hotspots related to daytime heat stress, as UTCI > 32 °C are rarely reached at night without solar radiation (Fig. 11b). Figure  shows the same map extent but with nighttime hours exceeding a UTCI value of 22 °C. Figure 9 and A2 provide a more detailed view of four exemplary urban areas representing the LCZs 2, 5, and 8. These areas show not only the thermal comfort difference between different LCZs but also the intra-LCZ variability (Fig. 9c and 9d). The four areas





**Figure 8: Map of Freiburg representing the average amount of daytime hours with a UTCI ≥ 32 °C per pixel (corresponds to strong and more heat stress). Note, coloring is in accordance with predicted UTCI quantiles. Spatial resolution of the map 1 x 1 m and time period for averages is 2019–2022. Rectangles (a)–(d) define areas shown in more detailed in Fig. 9, 11, and A2.**

represent the densely built-up city center with partly large trees but also large open and paved areas (a), an industrial area with

mostly paved surfaces and only a few trees (b), an old building district with a large and old tree stock (c), and a relatively new

district built 1995–2005 (d).

In Fig. 9, it is quickly apparent that there are large differences due to shading and tree canopy during the day and night.

Figure 9e and A2e show the distributions of the spatially distributed amounts of hours in Fig. 9a–d. The distributions confirm

the visual perspective. The industrial area (b) and the new building district (d) have a higher share of pixels, with more hours

exceeding the threshold of 32 °C and fewer hours exceeding the threshold of 22 °C during the night. In turn, the densely built-

up city center (a) and the old building district (c) have fewer hours ≥ 32 °C but more hours > 22 °C. Figure 11 shows the

summertime UTCI distributions during the day- and nighttime for these four areas. The distributions underline the visual

assumption that tree-covered and less-paved areas are less affected by thermal stress during the daytime, but have higher UTCI

values during nighttime. In addition, Fig. 11 shows the 80[th] percentiles of the different areas, ranging from 29 °C (old



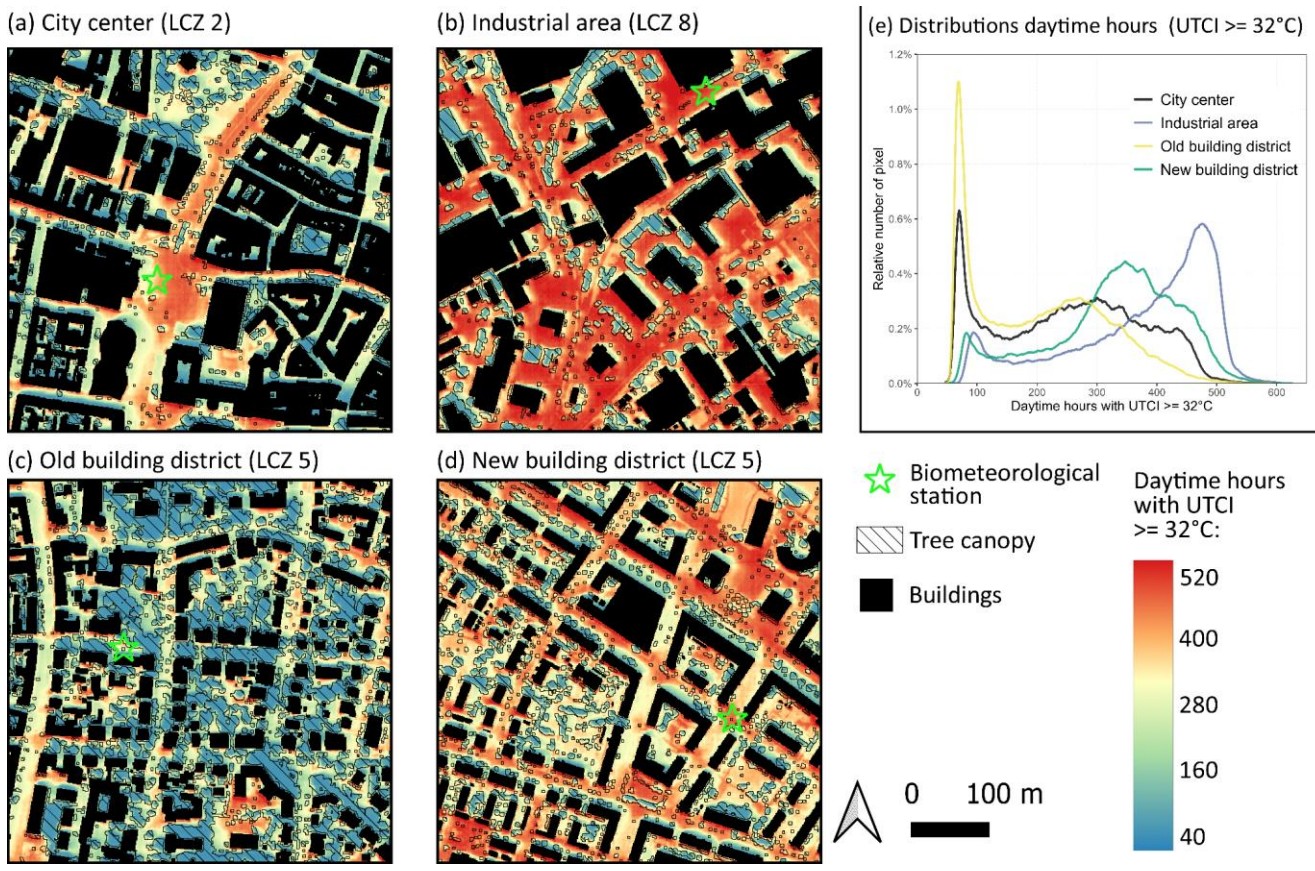

**Figure 9: Four 500 x 500 m subsets of different urban neighborhoods of Freiburg. Average hours per year with a UTCI ≥ 32 °C are shown. (a) shows the city center of Freiburg, (b) an industrial area in the north of Freiburg, (c) a district with old buildings and dense / mature tree stock, and (d) a building district built after the year 1995. (e) shows the distributions of (a)–(e). Note, coloring is in accordance with predicted UTCI quantiles. Spatial resolution is 1 x 1 m and time period is 2019–2022.**

building district) to 32 °C (industrial area) during the daytime and from 21 to 22 °C during nighttime. The 80[th] percentiles highlight the importance of shading to daytime outdoor thermal comfort. During the nighttime, however, the industrial area and the new building district cool down more quickly, resulting in lower UTCI values and 80[th] percentiles (21 °C vs 22 °C).

Although there is a cooling effect at night, the shading effect during the day is stronger than the reduction of nighttime cooling. These results indicate that outdoor thermal comfort controls have an inverted effect during the day and the night.

There are not only differences between the different LCZs but also within the same LCZ class. The old building and the new building district show strong differences in their day- and nighttime UTCI distributions. The difference between those two areas is larger than between the city center and the old building district or between the industrial and new building districts.

The areas do not differ much in their physical building characteristics and are both classified as LCZ 5. However, the old



**Figure 10: Map of Freiburg representing the average amount of nighttime hours with a UTCI ≥ 22 °C per pixel. Note, coloring is in accordance with predicted UTCI quantiles. Spatial resolution is 1 x 1 m and time period is 2019–2022. Rectangles (a)–(d) define areas shown in more detailed in Fig. 9, 11, and A2.**

building district has a tree cover fraction of 45 % and can be further defined as LCZ subclass $5_b$, while the new building district has only 24 % tree cover and can therefore defined as LCZ $5_d$.

In Fig. 10, sharp UTCI transitions are partially visible. These transitions occur between adjacent grid cells (500 x 500 m) of the $T_a$ and RH MLPs (= original SUEWS grid cells). The transitions are more prominent in Fig. 10 than in Fig. 8, as $T_{mrt}$ more influences diurnal UTCI than $T_a$. The transitions between grid cells also illustrate that $T_a$ and RH are modeled individually for each grid cell without interaction with adjacent cells.



**Figure 11: Summer time (May–September) distributions of UTCI of the four districts shown in Fig. 9 and A2 at daytime (a) and during nighttime (b). On top of each figure the UTCI heat stress classes are presented (CS: cold stress; HS: heat stress). Vertical lines in the colors of the four areas represent the 80th percentiles. Left y-axis shows the relative and right y-axis the absolute number of hours. Note: x-axis has a bin size of 1 °C; total amount of hours varies between (a) and (b) because of day length in summer.**

# 4    Discussion

## 4.1    Evaluation of the HTC-NN submodels

Both MLPs show promising results compared to measurements and especially compared to the numerical model SUEWS. With an overall (spatial) $R^2$ of 0.997 and an RMSE of 0.5 K, the $T_a$ MLP shows excellent accuracy. Both SUEWS and the $T_a$





MLP have an RMSE of around 1.5 K in relation to sensor network data, while the MAE is only at around 1 K. RMSE also varies with time (Fig. 4c). Apart from the two periods late October and mid-December, with RMSE values higher than 2 K,
RMSE fluctuates around 1 K for most of the remaining time. This suggests that the overall RMSE of 1.5 K is highly influenced by the two outlier periods in late October and mid-December with high RMSE values between 2 K and 3 K. For the RH MLP, similar observations are made, having higher RMSE in late October and mid-December. Since the MLPs and the SUEWS model have similar error patterns, the error must already be apparent in forcing data. The end of October was unusually warm, with clear sunny days but cold nights. As forcing data is measured on a rooftop at 55 m a.g.l., the near-surface night cooling
is not represented in this data, leading to higher RMSE values. In December, on the other hand, an uncommon weather event occurred with a surface inversion between the surface and the rooftop at 55 m a.g.l., with colder air near the surface and warmer air above. Both SUEWS and the MLP have a lower RMSE than the forcing data in relation to the sensor network data. However, the gain in accuracy is moderate for RH at 1 % and for $T_a$ at 0.3 K. One reason could be that SUEWS is set up in default mode, and no specific model calibrations are performed. In addition, the modeled soil moisture has not been validated
against measurement data. Another reason could be that SUEWS is run in offline mode. Running SUEWS in online mode, coupled to a mesoscale weather model would probably increase its accuracy further as boundary conditions and tiles interactions, including local wind systems, would be better mapped.

The RF approach to model a statistical wind field for the entire urbanized area of Freiburg shows good results for outdoor thermal comfort modeling. Since the RF models only a statistical wind field, not every wind gust can be accurately represented.
Still, as, the model is forced with hourly aggregated data, it is sufficient to predict hourly averaged $U$.

The results of the evaluation of the $T_a$ and RH MLPs and the RF illustrate the power and potential of deep and machine learning models in the context of urban climate when appropriate training data is available. It can also be seen, that the MLPs have similar model shortcomings as SUEWS, and an enhanced MLP performance requires an improved numerical model.

### 4.2   Evaluation HTC-NN

The tradeoff between computational cost and model accuracy of the HTC-NN compared to numerical models is positive for both. The HTC-NN has a higher model accuracy than SOLWEIG with its POI option. In addition, model accuracy is constant throughout the year, with lower accuracy during less common weather conditions. As mentioned, uncommon weather situations with strong surface inversions in the city are hard to predict for numerical and deep learning models. The overall RMSE of UTCI is around 3 K. Diurnal error patterns of UTCI show higher errors during the day than during the night due to
higher errors of $T_{mrt}$ predictions. However, the highest errors occur before noon, while in the late afternoon, UTCI predictions are more accurate (RMSE of about 3K) when thermal stress is highest. During the heat wave event, the HTC-NN overestimates UTCI to different degrees during the day. It is also apparent that daytime UTCI follows the patterns of $T_{mrt}$ most of the time, emphasizing the importance of correct shadow pattern data. Since $T_{mrt}$ predictions of SOLWEIG and HTC-NN are partly in line but UTCI values differ, the prediction error must be related not only to $T_{mrt}$ o $T_a$ and RH, while $U$ is negligible during these
days. These results indicate that the accuracy of the HTC-NN is affected to varying degrees by its submodels in different





weather conditions and that an overall attribution of error to the submodels cannot be made but must be done individually for the different weather conditions.

To the authors' knowledge, not many comparisons have been made between UTCI models and measurements with at least one week of comparison. Nice et al. (2018) used a modified version of the TUF-3D model (Krayenhoff and Voogt, 2007) to model UTCI in a suburb of Melbourne for four weeks. Observed UTCI values were calculated from $T_{mrt}$ (black globe), $T_a$, RH, and $U$ measurements. A total of seven stations were installed and compared to model results. MAE between modeled and observed UTCI ranged from 1.80 K to 3.03 K and RMSE from 2.33 to 3.64 K, which aligns with HTC-NN. On the other hand, Meili et al. (2021) applied the ecohydrological model UT&C in Singapore with a model accuracy for UTCI ranging from 1.9 to 3.1 K (RMSE). Since Freiburg is hardly comparable to Melbourne or Singapore, these findings still help to better evaluate the HTC-NN model results. The HTC-NN has a similar accuracy but allows for modeling larger domains on high resolution.

While the HTC-NN has a very good trade-off between accuracy and computational cost, it has some limitations. First, the HTC-NN is not coupled to a mesoscale model and thus does not include local or mesoscale weather phenomena, such as mountain valley wind systems, cold air drainage and advection of heat (e.g., heat island and urban plume). Additionally, while large model domains are possible, the 500 x 500 m model tiles for $T_a$ and RH are modeled individually, and no boundary or heat and moisture transport effects are considered, as in the offline SUEWS version. These constraints are related to the model structure of the offline SUEWS version and could be partially resolved by running it coupled to a mesoscale model. Another constraint is related to the forcing data. The HTC-NN is forced with meteorological data from a weather station at 55 m a.g.l. Measured $T_a$ at this height may already be exposed to other processes as near-ground level $T_a$, which may lead to initial error propagation. Additionally, the larger the model domain, the more difficult it is to force the model with data from one weather station. This could be solved by forcing the HTC-NN with reanalysis data or weather forecast products, which would reduce the model fit due to inconsistencies between measured and reanalysis and forecast data. As the submodels perform well in emulating the numerical models, better numerical models would also be needed to increase the accuracy of the model itself (Briegel et al. 2023).

The evaluation of the HTC-NN on sensor network data and the comparison with similar studies show that the HTC-NN is a valuable tool for modeling outdoor thermal comfort in complex urban areas.

## 4.3    High-resolution UTCI mapping

HTC-NN is used to predict UTCI for Freiburg for four years with high temporal (hourly) and spatial resolution (1 m). Almost 1.5 trillion predictions were made, which took around 8 days. To determine day and night heat hotspots, the hours above the specific 80th percentile (32 °C and 22 °C) are summed up and mapped. In addition, four specific areas representing LCZ 2, 5, and 8 are presented in more detail. The four areas differ in terms of their physical characteristics and also strongly in terms of their UTCI distributions. The daytime UTCI distributions of the new building district (d) and the industrial area (b) are flatter and shifted to higher values compared to the distribution of the city center (a) and the old building district (c). The 80th percentiles are shifted up to 3 °C. The new building district and the industrial area also have more pixels with UTCI values ≥





32 °C during the day. At night, however, it is the other way around, and the city center and the old building district have more
pixels with UTCI > 22 °C, which is also present in the nighttime UTCI distributions. This effect can be largely attributed to
the proportion of covered areas, either by tree canopy or dense building structure. The denser the buildings and the larger the
trees, the lower the sky view factor. Since $T_{mrt}$ is the driving factor for outdoor thermal comfort during the day, a reduced SVF
reduces the radiation load and, thus, the UTCI. At night, however, covered areas show reduced upward longwave radiation,
leading to higher UTCI values. These results indicate that adaption measures may work inconsistently during the day and at
night.

Besides the intra-urban variability of outdoor thermal comfort between different urban areas / LCZs, there is also an intra-
LCZ variability of outdoor thermal comfort. The old and new building districts have similar building characteristics but show
very different UTCI distributions for day and night. Intra-LCZ variability has the same magnitude as inter-LCZ variability, as
seen by their distributions and 80th percentiles. This is due to the remaining land cover characteristics besides building
structures such as tree or grass cover fraction. Outdoor thermal comfort has high variability at the microscale due to $U$ and
$T_{mrt}$, highlighting the importance of high-resolution modeling. These results further indicate that the urban climate
characteristics of a small city with large green areas, such as Freiburg, are better described by the LCZ subclasses or directly
by land cover class fractions.

## 5    Conclusions

This study presents a novel deep learning approach to outdoor thermal comfort modeling, the Human Thermal Comfort Neural
Network (HTC-NN). The HTC-NN consists of four submodels that separately model $T_a$, RH, $T_{mrt}$, and $U$, followed by UTCI
calculation. The submodels are trained on numerical model results, essentially emulating the numerical models through
machine learning methods. Each submodel and the final HTC-NN are validated separately with data from a dense sensor
network.

The research objective to develop and evaluate machine learning models emulating numerical urban climate models could
be achieved (i). In addition, the evaluation of the HTC-NN shows promising results (ii). Furthermore, we could show that the
HTC-NN has a positive trade-off between accuracy and computational cost. The accuracy of the HTC-NN is comparable to
numerical urban climate models (RMSE of 3 K to 3.3 K from SOLWEIG), while it is computationally superior. This
computational superiority allows high-resolution modeling of outdoor thermal comfort for large domains and long periods.
The HTC-NN is used to model UTCI for four years (2019–2022) for Freiburg with a temporal and spatial resolution of one
hour and 1 x 1 m to determine intra-urban variability of outdoor thermal comfort (research objective iii). The advantage of the
HTC-NN is that it is able to model outdoor thermal comfort with high spatial and temporal resolution, which allows the
investigation of spatial and temporal patterns of outdoor thermal comfort. Therefore, high resolution UTCI predictions can be
aggregated either temporally or spatially, or both. The HTC-NN further allows studying thermal comfort patterns only for
specific weather events, UTCI classes, or selected areas.





We demonstrate that HTC-NN is fast and versatile enough to continuously model long periods and entire cities using a building-resolved approach. For urban climate services, but also urban climate assessments and environmental consulting applications, there is no longer a need to base assessments on numerical simulations of a few selected "case studies", but instead we can build entire climatologies of thermal comfort, including the corresponding exceedance frequencies, explicitly
on end-user machines.

Nevertheless, the HTC-NN has limitations, and its applicability to other cities needs further investigation. It does not consider the city surroundings and its mountains with elevation differences up to 1000 m, as it is not coupled to a mesoscale weather model. This means it does not consider local wind systems, shading by hills, or interactions between different local scale grid cells (500 x 500 m). In addition, heat advection is not taken into account. Regarding the transferability to other cities, as long
as the city has similar building and vegetation characteristics, it could be easily transferred since the training data covers a wide range of urban structures. However, mapping unknown urban forms, such as skyscrapers or denser building structures, is critical, as they are not present in the training data. Nevertheless, it could be applied to unknown cities after validation of the trained submodels by numerical models or on measured data.

Although only one potential application of the HTC-NN has been considered in the present paper, several applications are
enabled by the low computational cost of the HTC-NN. These applications include "simple" urban-specific thermal comfort predictions and warnings for the next days using weather forecast models. The HTC-NN, however, could also be applied to the downscaling of potential future climates to building scale (e.g., using EURO-CORDEX data), or the assessment of the effectiveness of adaptation measures by changing the input data on urban form. Finally, HTC-NN can be used to investigate the driving forces of thermal comfort at different scales and hence fundamentally develop guidelines in support of urban
planning and policymaking.

**Code and data availability**

The code of the MLPs, the U-Net, and the UTCI calculation are available at https://doi.org/10.5281/zenodo.7974472. However, the digital elevation model data are released by the city of Freiburg only on a restricted basis. Nevertheless, they can be requested for scientific purposes from the Surveying Office of the city of Freiburg. All spatial and meteorological data and
model results can be found at https://doi.org/10.5281/zenodo.7974307.

**Author contributions**

FB designed the HTC-NN and prepared the original draft in cooperation with AC. JW designed the RF wind model. FB analysed and visualised the results. AC and DS provided supervision and review of the original draft.



## Competing interests

The authors declare that they have no conflict of interest.

## Appendix A

**Table A1: Validation measures of the machine learning models against the numerical models. In contrast to 4, where only pointwise comparisons are made, this table shows mean, MAE, MBE, RMSE, and $R^2$ for the entire test areas (spatial comparison) for 2022.**

| Variable | Model combination | Mean | MAE | MBE | RMSE | $R^2$ |
|---|---|---|---|---|---|---|
| $T_a$ | SUEWS / MLP | 13.36 °C / 13.38 °C | 0.31 K | -0.02 K | 0.50 K | 0.997 |
| RH | SUEWS / MLP | 70.43 % / 70.69 % | 2.14 % | -0.26 % | 3.28 % | 0.97 |
| $U$ | LES / RF | 1.12 m s$^{-1}$ / 0.58 m s$^{-1}$ | 0.55 m s$^{-1}$ | -0.54 m s$^{-1}$ | 0.76 m s$^{-1}$ | 0.30 |



**Figure A1: Modeled and measured UTCI at five Tier I stations (a)–(e) during a cold wave in December 2022.**






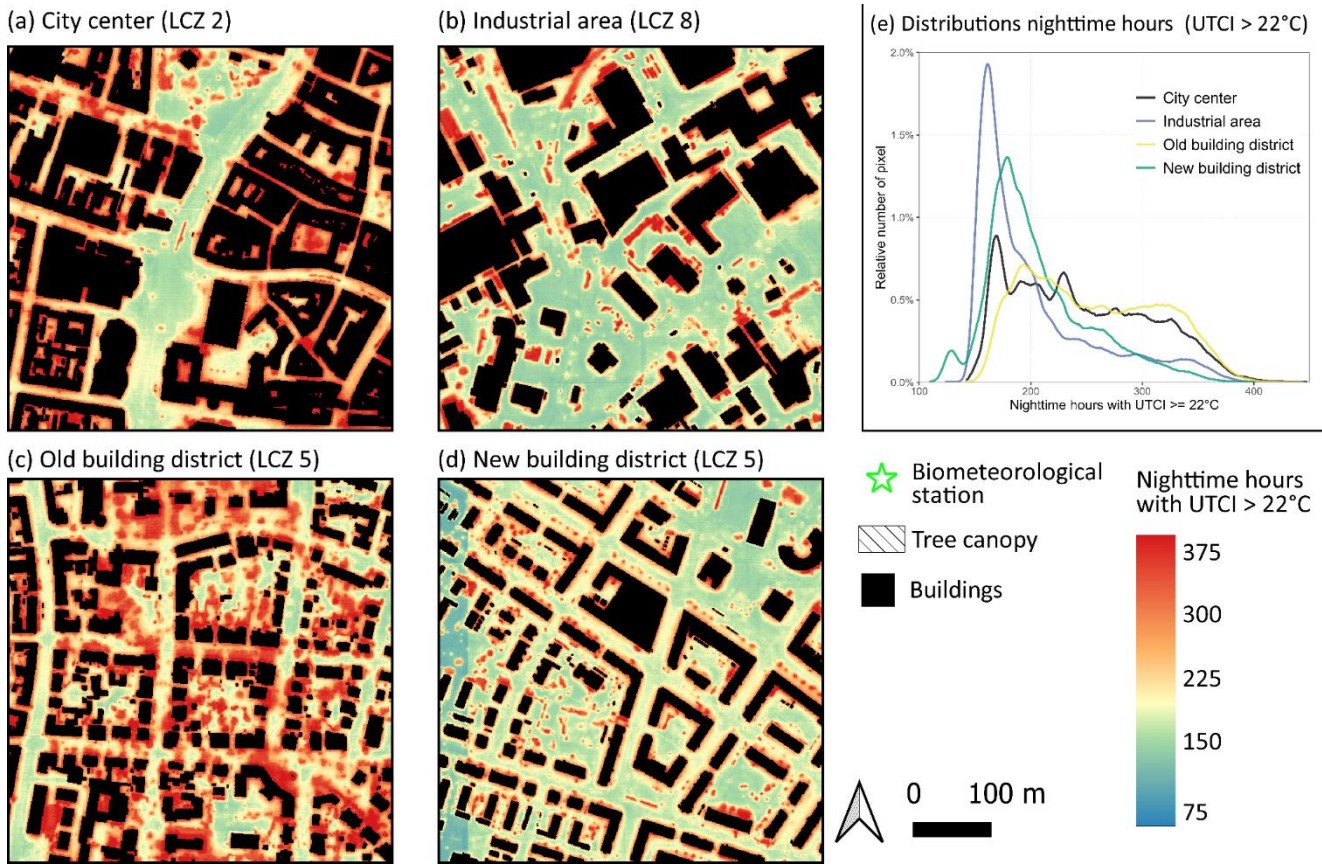

**Figure A2: Four 500 x 500m subsets of different urban neighborhoods of the city of Freiburg. Average nighttime hours with a UTCI ≥ 22 °C are shown. (a) shows the city center of Freiburg, (b) an industrial area in the north of Freiburg, (c) a district with old buildings and dense / mature tree stock, and (d) a building district built after the year 1995. (e) shows the distributions of (a)– (e). Note, coloring is in accordance with predicted UTCI quantiles. Spatial resolution is 1 x 1m and time period is 2019–2022.**

**Acknowledgments**

The model development and evaluation were funded by the German Federal Ministry for the Environment, Nature Conservation and Nuclear Safety (BMU) on the basis of a resolution of the German Bundestag as part of the 'KI-Leuchtturm'
project 'Intelligence for Cities' (I4C). Validation data (sensor network) used in this research were collected as part of the ERC Synergy Grant 'urbisphere' project, funded by the European Research Council (ERC-SyG) within the European Union's Horizon 2020 research and innovation program under grant agreement no. 855005. Spatial data (DEM / DSM) was provided by the administration of the city of Freiburg. We gratefully acknowledge Matthias Zeeman, Marvin Plein, and Gregor Feigel from the University of Freiburg for the installation of the sensor network, the data management, and for providing the data and
Marco G. Giometto from Columbia University for assisting with the Large Eddy Simulations.



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
