# Peer review of "High-resolution multi-scaling of outdoor human thermal comfort and its intra-urban variability based on machine learning"

_Geoscientific Model Development, 2023_

## Referee Comment (RC1)

General comment:

The manuscript addresses the need for accurate thermal comfort mapping in urban environments, which increasingly experience heat waves. Here, the authors present the approach they have developed: the Human Thermal Comfort Neural Network (HTC-NN). The modeling approach demonstrates a comprehensive integration of numerical models and machine learning techniques to predict human thermal comfort in high resolution urban environments. Evaluation is done for Freiburg, Germany in predicting street-level Universal Thermal Climate Index (UTCI) with high spatial resolution using street-level measurements. Variations in thermal comfort and hot spot disparities during different times of the day are identified at the neighborhood-level scale.

Overall, the manuscript provides a well-structured overview of the study. I have enjoyed reviewing it and I congratulate the authors with this work: it makes for an impressive contribution to the field of urban climate research; however I do have some remarks for improvement mainly in the evaluation section. I recommend it is accepted with minor revisions as detailed below.

Specific comments:

**Section 2.4**

Here it would be nice to have some absolute statistics about the reference period, to provide some context. What is mean JJA temperature (or annual cycle) over the study area, quantify the heat extremes in this period (how many, what temperature?), etc from ERA5.

**Section 2.5**

The method is quite complex. It is a very important part of this manuscript as this is a development/technical paper. There can be more structure in the modeling approach section, and in the section of each submodel's development.

The way I understand it now is the following: The development of HTC-NN involves four main steps: First, initial spatial and meteorological data are generated from various sources. Then, so-called "ground truth" data for the four HTC-NN submodels (MLPs, U-Net, RF) is calculated using numerical models. This includes data for air temperature (Ta), relative humidity (RH), mean radiant temperature (Tmrt), and wind speed (U). The third step is the training and evaluation of the submodels, which include two Multi-Layer Perceptrons (MLPs) for modeling Ta and RH at the neighborhood scale, U-Net for modeling Tmrt at the building-resolved scale (that has already been trained and validated in an earlier study) and a Random Forest (RF) model for statistical wind fields. The final step involves linking the submodels and calculating the Universal Thermal Climate Index (UTCI).

In particular I suggest the authors to make this section more explicit as to what parts are physically modelled, and what parts are trained submodels. It is physical modelling part of the HTC-NN? Line 141 is confusing to me; I am expecting 3 subsections in 2.5 (i.e., two MLPs and RF), but there are 4. Further, I think it will be beneficial to explicitly state per submodel what it takes as input and forcing data. Regarding the physical modelling/preprocessing; consider

adding an extra subsection to section 2 where you can explain how you have used LES and SUEWS.

**Section 2.6**
Please expand the explanation of UTCI, elaborate and provide the definition of the heat stress classification groups. Later in your analysis you use these terms: strong, very strong, or extreme heat stress (e.g., l.279).

**Section 3.1 and 3.2**
Looking at figure 1, your sensor data is mainly situated in urban sites, while your model area has a considerable fraction of more open fields. That may skew your observations. Please validate the Ta, RH, and U submodel components as well as the UTCI temperature with ERA5 and/or other types of reanalysis data full training period (2018-2022) . That will clarify whether the errors you find, such as the peaks in October and December (l.220), are robust.

**Section 4**
In the discussion section, the authors thoroughly discuss the model's performance and limitations. It is highlighted that HTC-NN demonstrates a favorable balance between computational cost and accuracy compared to numerical models. Remarkable errors are discussed, among which diurnal error patterns impacting UTCI predictions and the model's tendency to overestimate UTCI during the day during heatwaves. Limitations include the absence of coupling with a mesoscale model, neglecting local weather phenomena, and individual modeling of Ta and RH tiles. Dependency on data from a single weather station and potential initial error propagation are acknowledged issues. Suggestions for improvement involve coupling with mesoscale models and using reanalysis data.

Can you elaborate more generally on the limit of NNs, particularly training a network with a limited amount of extreme events?

**Figure 1:**
I find the gray grid cells in the figure are not well visible. Perhaps you can experiment with a different shade of gray, or explicitly mention in the caption something along the lines of "Note: Gray grid cells indicating the training areas of the Ta and RH submodels may be less visible due to color contrast."

**Table 1 and 2:**
Please write out the used abbreviations (LCC, DEM etc.) at their first use.

---

## Referee Comment (RC2)

The manuscript presents the results of modeling bioclimatic conditions in Freiburg. The authors compare the results obtained using numerical models with the results of models using machine learning techniques. In order to determine the spatiotemporal variability of the UTCI index, four ML models emulating appropriate numerical models are used. It was shown that properly trained ML models give results comparable to those of numerical models at much lower computational costs. The approach of using several independent numerical models with different resolutions to obtain the variables necessary to determine UTCI and then emulating them with ML models, so that calculations of long-term variability can be performed in a reasonable time, may constitute a contribution to the development of bioclimatic research methodology in urban areas. From a purely bioclimatic point of view, an important result are high-resolution maps of exceeding specific UTCI thresholds determined on the basis of modeling results with a 1-hour time step for a four-year observation period. The manuscript is fully in line with the scope of the GMD journal, its layout is typical of scientific articles, the argument is logical and contains all the necessary information. As I am not a native speaker, I do not evaluate the correctness of the language, but I did not notice any incorrectness. For the above reasons, I believe that it can be published as is, with only very minor corrections.

Specific comments

Ln. 61 The authors state that "*Nevertheless, the emulated ML model can never exceed the accuracy of the numerical model because it is trained based on the model's results*", but in the results and discussion it turns out that the proposed ML models often give better results than numerical models. I propose to explain this contradiction.

Ln. 72 … *four cardinal wind directions* … – I am used to analyzing the wind field as three-dimensional. Did I misunderstand something?

Ln. 213 "*The error distributions of SUEWS and the MLP across the different stations are similar (Fig. 3a)…*" – I think that Fig. 3a shows the error for all stations rather than the error distributions across stations.

Ln. 227 "*The $T_{mrt}$ U-Net has a slightly lower accuracy than SOLWEIG (RMSE of 6.18 K to 5.86 K; R2 of 0.84 to 0.86)*" (and also in ln. 239) – the acceptable level of accuracy is usually a subjective choice. However, for $T_{mrt}$ in the standard ISO7726 (ISO, 1998), ISO recommends that the error in $T_{mrt}$ estimates should be within ±5°C. Could you please address/discuss this.

Fig. 4 The RMSE of SUEWS predictions (orange lines) are almost invisible - could they be bolded? Please change "Dez" to "Dec".

Fig. 6 Some of the sharp drops in $T_{mrt}$ and UTCI in the SOLWEIG charts (e.g. afternoon 2022-02-11 at the station marked "e") are likely the results of shading, which are directly calculated by SOLWEIG at ground level, while the reference data is from 3.5 m. Similarly at early morning or afternoon at other stations. Am I right? Anyway, could you comment on

these rapid, sawtooth changes in the SOLWEIG charts and their effect on the accuracy statistics.

---

## Author Comment (AC1)

**Reply on RC1 'Comment on gmd-2023-122'**

We are pleased that you recommend to publish our manuscript and we would like to thank you for your comments which helped us to better structure the manuscript but also to clarify potential misconceptions and the fact of the non-capability of ANNs to extrapolate.

**Comment 1:**
"*Section 2.4*
*Here it would be nice to have some absolute statistics about the reference period, to provide some context. What is mean JJA temperature (or annual cycle) over the study area, quantify the heat extremes in this period (how many, what temperature?), etc from ERA5.*"
**Response 1:**
We added absolute statistics to give a better overview of the study periods. We added information on summer mean temperature, average of daily maximum values, and number of heat days (maximum $T_a$ >= 30°C):

During the study period, the mean annual $T_a$ is 13.0°C, the mean summer (June-August) $T_a$ is 21.3°C, and the mean maximum daily summer $T_a$ is 26.3°C. The number of hot days (maximum $T_a$ >= 30°C) of the consecutive years from 2019 to 2022 are 26, 20, 9, and 37, respectively.

**Comment 2:**
"*… In particular I suggest the authors to make this section more explicit as to what parts are physically modelled, and what parts are trained submodels. It is physical modelling part of the HTC-NN? Line 141 is confusing to me; I am expecting 3 subsections in 2.5 (i.e., two MLPs and RF), but there are 4. Further, I think it will be beneficial to explicitly state per submodel what it takes as input and forcing data. Regarding the physical modelling/preprocessing; consider adding an extra subsection to section 2 where you can explain how you have used LES and SUEWS.*"
**Response 2:**
1. We re-arranged the structure of the method section. We divided the method section into two sections: a data section and a model section (*2 Data / 3 Modelling approach*). In the data section we give an overview of the study area (*2.1 Study Area*), the spatial and meteorological forcing data (*2.2 Spatial and forcing data*), the validation data (*2.3 Validation data*), and the study period (*2.4 Study period*). The modelling section on the other hand was re-arranged into a numerical modelling and a machine learning modelling part (*3.1 Numerical Modelling / 3.2 Machine learning modelling*). The section 3.1 covers the SUEWS and LES modelling (*3.1.1 Local scale $T_a$ and RH modeling (SUEWS) / 3.1.2 Micro-scale U modeling (LES)*), while the section 3.2 covers the model development of the two MLPs for modelling $T_a$ and RH (*3.2.1 Multi-layer perceptron model development*) and the random forest for modelling $U$ (*3.2.2 Random Forest model development (U)*). We hope that this re-arrangement clarifies the entire model development process.

2. In section *2.2 Spatial and forcing data* we mention which forcing data is used for the different numerical and machine learning models:

> The following variables are used as forcing data for SUEWS, the corresponding MLPs, and the U-Net: $T_a$, RH, atmospheric pressure, downwelling shortwave radiation, downwelling longwave radiation, precipitation, $U$ and wind direction. LES and the RF model requires only standard forcing related to an initial shear velocity of 1 m s-1.

However, we added some clarifications to section 3 and refer to table 2 where we added the forcing data:

> The development of the HTC-NN requires four steps (Fig. 2). The first step is to generate initial spatial and meteorological data from various sources, which are listed in Table 2. In the second step, the so-called 'ground truth' data (Ta, RH, Tmrt, and U) for the four HTC-NN submodels (two MLPs, U-Net, and RF) are calculated using numerical models (SUEWS, SOLWEIG, and LES). Training and evaluation of the HTC-NN submodels are done in the third step, while the fourth step is to link these submodels by calculating UTCI. As the U-Net has already been trained and validated, only the development and the requirements of the MLPs and the RF (spatial and temporal data, SUEWS, and LES) are explained.

**Table 2: Overview of required spatial and forcing data for the numerical and machine learning models. Note: SUEWS and MLP use abstract spatial data (see Table 1) and the RF model uses additional spatial predictors derived from DEM, DSMb, and DSMv which are not listed here.**

| Data | SUEWS / MLP (500 x 500 m) | SOLWEIG / U-Net (1 x 1 m) | LES / RF (1 x 1 m) |
|---|---|---|---|
| LCC map | x | x | - |
| DEM | x | x | x |
| DSMb | x | x | x |
| DSMv | x | x | x |
| Sky view factor | - | x | - |
| Wall height and aspect | - | x | - |
| Soil characteristics | x | - | - |
| *U* | x | x | x |
| $T_a$, RH, atmospheric pressure, downwelling shortwave radiation, downwelling longwave radiation, precipitation, wind direction | x | x | - |

**Comment 3:**

*"Section 2.6

*Please expand the explanation of UTCI, elaborate and provide the definition of the heat stress classification groups. Later in your analysis you use these terms: strong, very strong, or extreme heat stress (e.g., l.279).”*

**Response 3:**

We state out that UTCI can be categorized into different thermal comfort classes. We also added a table to Appendix A (Table A1), where all UTCI classes and the corresponding thermal comfort classes are listed:

The UTCI values can be categorized based on thermal stress, e.g., UTCI values ranging from 32–36°C are assigned to strong heat stress. The different UTCI stress categories and the corresponding UTCI ranges are listed in Table A1.

**Table A1: Universal thermal climate index (UTCI) classification of thermal stress (Błażejczyk et al., 2013).**

| UTCI (°C) | Stress category |
|---|---|
| > +46 | Extreme heat stress |
| +38 – +46 | Very strong heat stress |
| +32 – +36 | Strong heat stress |
| +26 – +32 | Moderate heat stress |
| +9 – +26 | No thermal stress |
| 0 – +9 | Slight cold stress |
| -13 – 0 | Moderate cold stress |
| -27 – -13 | Strong cold stress |
| -40 – -27 | Very strong cold stress |
| < -40 | Extreme cold stress |

**Comment 4:**

“*Looking at figure 1, your sensor data is mainly situated in urban sites, while your model area has a considerable fraction of more open fields. That may skew your observations. Please validate the Ta, RH, and U submodel components as well as the UTCI temperature with ERA5 and/or other types of reanalysis data full training period (2018-2022) . That will clarify whether the errors you find, such as the peaks in October and December (l.220), are robust.*”

**Response 4:**

We have made an additional comparison of the results from SUEWS and the $T_a$ and RH MLPs to investigate whether the error peaks in October and December are related to the forcing data or to the MLP models. A direct comparison between modelled $T_{mrt}$ and $U$ values with ERA5 Land data is difficult due to the spatial variability of $T_{mrt}$ and $U$ at the micro-scale, which tend to dominate the global effects of the forcing data. Nevertheless, we will include the results of the $T_a$ / RH comparison in the appendix of our manuscript (see figure 1 in this document). This figure shows that the error peaks in late October and mid-December are caused by the forcing data and forwarded to the model data. We mention this also in the discussion:

Compared to the ERA5 land data, the forcing and model data show higher errors during these periods in October and December, indicating that errors are already present in the forcing data and are passed on to the model results (Figure A1).

[Figure]

**Figure 1: Moving average of RMSE of $T_a$ (a) and RH (b) from August to December 2022. As reference data ERA5-Land data is used (Muñoz Sabater, 2019). The window size of the moving average is seven days. Time series starts with the installation of the first Tier I stations in August 2022. Shaded areas represent 95 % confidence interval.**

**Comment 5:**

*"Can you elaborate more generally on the limit of NNs, particularly training a network with a limited amount of extreme events?"*

**Response 5:**

We added a discussion on ANN an its dependencies on the occurrence of extreme events or spatial structures within the training data. We made a statement, that ANN are able to interpolate but nor to extrapolate which should clearly underline the fact, that ANN should be treated with caution when applied to cases with rare training data:

Nevertheless, the HTC-NN should only be applied to 'known' spatial and temporal data, as ANN are generally capable of interpolation but not extrapolation. This means that similar urban structures and/or meteorological forcing data are suitable as potential prediction data. However, any unknown spatial configurations or unknown extreme weather events, should

be approached with caution and undergo validation against measurement or numerical model data.

**Comment 6:**

*"Figure 1:*

*I find the gray grid cells in the figure are not well visible. Perhaps you can experiment with a different shade of gray, or explicitly mention in the caption something along the lines of "Note: Gray grid cells indicating the training areas of the Ta and RH submodels may be less visible due to color contrast."*

**Response 6:**

We changed the shade of gray. It is now brighter and should be better visible:

[Figure]

**Figure 2: Model domain of the City of Freiburg, Germany. The red star shows the location of the weather station used for forcing data on a rooftop. Orange and turquoise points show the locations of the urban sensor network used for model evaluation. Gray grid cells show the training areas of the $T_a$ and RH submodels, while yellow grids show the test areas. Red and blue squares show the training and test areas of the $U$ sub model, respectively. The pink border shows the prediction area of UTCI. Orthophoto in the background based on data from the City of Freiburg, www.freiburg.de.**

**Comment 7:**

*"Table 1 and 2:*

*Please write out the used abbreviations (LCC, DEM etc.) at their first use."*

**Response 7:**

We added the long names to the captions.

**References**

Błażejczyk, K., Jendritzky, G., Bröde, P., Fiala, D., Havenith, G., Epstein, Y., Psikuta, A., & Kampmann, B. (2013). An introduction to the Universal Thermal Climate Index (UTCI). In *Geographia Polonica* (Vol. 86, Issue 1). IGiPZ PAN. http://rcin.org.pl/igipz/Content/29010/WA51_46784_r2013-t86-no1_G-Polonica-Blazejcz1.pdf

Muñoz Sabater, J. (2019). *ERA5-Land hourly data from 1981 to present. Copernicus Climate Change Service (C3S) Climate Data Store (CDS). (Accessed on < 21-04-2022 >)*. 10.24381/Cds.E2161bac.

---

## Author Comment (AC2)

**Reply on RC2 'Comment on gmd-2023-122'**

We are pleased that you are in favour of our manuscript and we would like to thank you for your suggestions for some corrections.

**Comment 1:**

"*Ln. 61*

*The authors state that 'Nevertheless, the emulated ML model can never exceed the accuracy of the numerical model because it is trained based on the model's results', but in the results and discussion it turns out that the proposed ML models often give better results than numerical models. I propose to explain this contradiction.*"

**Response 1:**

Indeed. In some situations, the ML models are more accurate than the numerical models. There are several reasons for this. First, the UTCI estimated by the ML model is closer to the observations because it combines several downscaling models ($T_{mrt}$, $T_a$, RH, and wind speed) that take urban form and function into account. SOLWEIG, on the other hand, does not downscale $T_a$, RH, and wind speed. Another reason is simply luck. An ML model with high accuracy compared to a numerical model sometime has the error on the 'right side', which may lead to slightly higher accuracy when compared to observations. Nevertheless, we added a clarification to the manuscript (Line 344):

The lower RMSE of the HTC-NN compared to SOLWEIG can be explained by the combination of four submodels that downscale $T_a$, RH, $T_{mrt}$, and $U$ separately, while SOLWEIG downscales only $T_{mrt}$ comprehensively.

**Comment 2:**

"*Ln. 72*

*'… four cardinal wind directions …' – I am used to analyzing the wind field as three-dimensional. Did I misunderstand something?*"

**Response 2:**

Thank you for this comment. We have calculated the wind speed from the x, y, and z components. The four cardinal wind directions only describe the general inflow direction of the LES and ML models. That is, we computed statistical wind fields using x, y, z components, but only for four wind directions due to computational cost. We added the following sentence at line 72:

The wind fields are calculated from the x, y, and z wind components.

**Comment 3:**

"*Ln. 213*

*'The error distributions of SUEWS and the MLP across the different stations are similar (Fig. 3a)…' – I think that Fig. 3a shows the error for all stations rather than the error distributions across stations.*"

**Response 3:**

You are right. Fig. 3 does not show error distributions of the models but boxplots. The boxplot shows all errors of all stations. We changed this in the text (line 221):

The errors  of SUEWS and the MLP across the different stations are similar (Fig. 3a), with a higher variability during the night than the during day.

**Comment 4:**

*"Ln. 227*

*'The $T_{mrt}$ U-Net has a slightly lower accuracy than SOLWEIG (RMSE of 6.18 K to 5.86 K; R2 of 0.84 to 0.86)' (and also in ln. 239) – the acceptable level of accuracy is usually a subjective choice. However, for $T_{mrt}$ in the standard ISO7726 (ISO, 1998), ISO recommends that the error in $T_{mrt}$ estimates should be within ±5°C. Could you please address/discuss this."*

**Response 4:**

You are right. ISO7726 recommends that measurements or models should estimate $T_{mrt}$ within ±5°C. On average we are indeed within ±5°C, as the mean absolute errors (MAE) are 4.25 K and 3.83 K for the ML model and SOLWEIG respectively. However, the root mean square error is higher as this error metric gives more weight to outliers. Due to complex shadow patterns it is very difficult to always predict $T_{mrt}$ accurately, even for numerical models (e.g., see also Fig. 4 (b) and Briegel et al. 2022). It is inevitable to always model $T_{mrt}$ within the ±5°C range. Therefore, for the purpose of modelling outdoor urban thermal comfort, we believe that the achieved model accuracy of the ML is sufficient.

**Comment 5:**

*"Fig. 4*

*The RMSE of SUEWS predictions (orange lines) are almost invisible - could they be bolded? Please change "Dez" to "Dec"."*

**Response 5:**

Thank you for this suggestion. We changed it. The orange lines showing the SUEWS results are still not perfect to see. That is because the errors of SUEWS and the ML model are almost in line which makes it hard to make the both error lines perfectly visible.

[Figure]

**Comment 6:**

"*Fig. 6*

*Some of the sharp drops in $T_{mrt}$ and UTCI in the SOLWEIG charts (e.g. afternoon 2022-02-11 at the station marked "e") are likely the results of shading, which are directly calculated by SOLWEIG at ground level, while the reference data is from 3.5 m. Similarly at early morning or afternoon at other stations. Am I right? Anyway, could you comment on these rapid, sawtooth changes in the SOLWEIG charts and their effect on the accuracy statistics.*"

**Response 6:**

This is actually a very good point. The complex shadow patterns within the city as well as the different heights between model and observations could be reasons for the rapid and strong changes in the $T_{mrt}$ and UTCI predictions. We have not investigated the impact of these strong outliers on the overall accuracy of the model. However, your question is also related to comment 4: RMSE are higher than MAE by 1-2 K. The reason for this could be the strong outliers you mention in this question. The MAE of SOLWEIG ranges from 1.99 to 3.30 K and the MAE of HTC-NN (U-Net) from 1.74 to 3.28 K for the different sensor stations. The plot you mention (Fig. 6 (e)) has actually the highest overall MAE of all stations with 3.30 K, while Fig. 6 (b) has the lowest with 1.99 K. Fig. 6 (b) does not show any rapid, sawtooth changes.

So, we can conclude that a more detailed investigation of the error patterns would be beneficial for further research and model development. But as we don't intend to evaluate numerical models in this study this would exceed the scope of this research.